# Immune Dysregulation Connecting Type 2 Diabetes and Cardiovascular Complications

**DOI:** 10.3390/life15081241

**Published:** 2025-08-05

**Authors:** Katherine Deck, Christoph Mora, Shuoqiu Deng, Pamela Rogers, Tonya Rafferty, Philip T. Palade, Shengyu Mu, Yunmeng Liu

**Affiliations:** Department of Pharmacology and Toxicology, University of Arkansas for Medical Sciences, Little Rock, AR 72205, USA; kdeck@uams.edu (K.D.); cjmora@uams.edu (C.M.); sdeng@uams.edu (S.D.); prkitkat@gmail.com (P.R.); raffertytonyam@uams.edu (T.R.); ppalade@uams.edu (P.T.P.); smu@uams.edu (S.M.)

**Keywords:** immune cells, cardiovascular diseases, Type 2 diabetes, insulin, leptin, adiponectin, IFNγ, TNFα, IL17, IL6

## Abstract

Type 2 diabetes (T2D) is a prevalent metabolic disorder characterized by persistent hyperglycemia, hyperinsulinemia, and long-term cardiovascular complications. Another hallmark of T2D is disrupted hormonal homeostasis—marked by elevated levels of insulin and leptin and reduced adiponectin—which plays a crucial role in modulating immune cell function. Individuals with T2D exhibit a skewed immune profile, with an elevated secretion of pro-inflammatory cytokines such as IFN-γ, TNF-α, IL17, and IL6, which are well-established drivers of vascular inflammation and dysfunction. Moreover, dysregulated metabolic hormones in T2D promote the acquisition of a pro-inflammatory phenotype in immune cells, suggesting that these hormones not only regulate energy balance but also serve as potent immune activators. Their dysregulation likely plays a significant—and perhaps underappreciated—role in the onset and progression of diabetic cardiovascular complications.

## 1. Introduction

T2D is initially characterized by prolonged elevated blood glucose and often co-exists with a cluster of clinical features known as metabolic syndrome [1]. These features encompass central obesity, hyperlipidemia, hyperglycemia, and hyperinsulinemia [2]. According to the latest estimates by the International Diabetes Federation, approximately 463 million individuals worldwide, including 38.4 million adults in the United States alone, are living with diabetes in 2024, with T2D accounting for about 90% of cases [3]. Furthermore, individuals with T2D face a 2- to 4-fold higher risk of developing cardiovascular diseases (CVDs), which include coronary heart disease, cerebrovascular disease, peripheral artery disease, and aortic atherosclerosis [3]. Among T2D patients, 40–50% exhibit chronic kidney disease (stage 1–4) and 80% have hypertension [4,5,6,7]. These conditions and diseases collectively contributed to the occurrence of heart failure and hemorrhagic stroke, which stand as the leading causes of mortality in individuals with type 2 diabetes [8,9].

The escalating prevalence of T2D is closely linked to the rise in rates of obesity, primarily fueled by overeating and sedentary lifestyles [10,11]. Obesity serves as a significant predictive risk factor for T2D development, highlighting the critical role of lifestyle factors in disease prevention and management [12]. Additionally, hyperglycemia emerges as a hallmark clinical feature of T2D, triggering protein hyper-glycosylation and enhanced oxidative stress, which in turn lead to cellular and systemic dysfunction, contributing to the detrimental physiological outcomes of T2D [13,14]. Although combining multiple glucose-lowering agents significantly improves glucose management for T2D patients, patients undergoing intensive glucose treatment experience higher cardiovascular mortality than those receiving conventional therapy [15,16,17]. This suggests the presence of unknown pathogenic factors contributing to the increased cardiovascular risk in T2D and highlights the urgent need to identify intrinsic factors that cause CVD in T2D patients [18,19,20,21,22,23,24].

Heightened levels of cytokines—long recognized as pathogenic factors sustaining chronic inflammation—contribute to the progression of tissue injury and the systemic manifestations of cardiovascular complications, including atherosclerosis, hypertension, coronary heart disease, and heart failure [25]. Immune cells, as the primary producers of cytokines, have emerged as indispensable players in inducing T2D-associated multiple complications through excessive production of pro-inflammatory cytokines [26]. Notably, the expression of hormonal receptors, including the InsR (insulin receptor), leptin receptor (LR), and adiponectin receptor (ADR) on immune cells [27,28,29], introduces an additional layer of regulation. This hormone-immune interface may represent an overlooked pathogenic pathway in driving the onset of CVDs in T2D.

### 1.1. Dysregulated Insulin/InsR Signaling and Its Implications in the Development of T2D and CVDs

Insulin, an important hormone developed during evolution, plays a central role in storing nutrients within our body cells to mitigate food scarcity [30]. Upon the breakdown of food into small molecules such as glucose by the digestive system, plasma glucose levels reach a threshold that triggers insulin release from pancreatic β-cells [31]. Released insulin exerts its effects by binding to insulin receptors (InsR) on the target cells [32]. First, it crucially regulates glucose availability inside the cell. This phenomenon was elucidated in immune cells through the radio-labeled insulin technique in the 1970s [33,34]. Additionally, its primary anabolic effect includes promoting glycogen and protein synthesis while inhibiting lipolysis in metabolic tissue such as the liver, skeletal muscle, and adipose tissue [35]. In times of nutrient surplus, beta cells release excessive amounts of insulin to store glucose as glycogen in the body [36]. However, prolonged nutrient surplus eventually results in the dysregulation of insulin signaling, commonly known as “insulin resistance”. This dysfunction contributes to systemic metabolic disturbances, including hyperglycemia, hyperinsulinemia, and hyperlipidemia in obese T2D patients [37]. Notably, “insulin resistance” involves both decreased and preserved insulin actions in a tissue-specific manner [38,39]. Specifically, suppressed insulin signaling in hepatocytes results in a limited uptake of glucose and enhanced gluconeogenesis, which contributes to hyperglycemia [40]. In contrast, insulin-sensitive hepatocytes engage in lipogenesis through insulin-induced transcriptional activation of SREBP-1(sterol regulatory element-binding protein), leading to systemic hyperlipidemia [41].

In the context of vascular biology, insulin leads to cardiovascular impairment. A recent clinical study reported that insulin prescription in T2D patients was associated with a higher risk of all-cause mortality [42,43]. In animal models, chronic insulin administration was found to increase mean arterial blood pressure [44,45,46]. Additionally, intravenous insulin infusion in healthy volunteers led to a 50% reduction in urinary sodium excretion [47]. Given that sodium reabsorption is paramount for regulating blood volume and pressure [48], chronic hyperinsulinemia-mediated sodium retention may contribute to the development of hypertension in T2D patients. While insulin was also known to promote vasodilation through enhancing endothelial NO (nitric oxide) production [49], the long-term vascular effects of hyperinsulinemia remain unclear. Further mechanistic studies are needed to elucidate how chronic hyperinsulinemia contributes to cardiovascular complications in T2D, which could inform the development of more effective treatment strategies.

With the growing understanding of the role of immune cells in T2D and CVD, the expression of InsR on these immune cells has garnered interest. Naïve T cells initially display relatively low expression of InsR, and their membrane InsR levels increase following T cell priming or activation [34]. This upregulation can stimulate their proliferation and enhance the production of IFNγ [50,51]. Conversely, CD4+ T cells deficient in InsR exhibited reduced IFNγ and proliferation upon antigen stimulation. Furthermore, exogenous insulin boosted the ECAR (extracellular acidification rate) and OCR (oxygen consumption rate), which indicated increased ATP production to support the demands for T-cell activation and effector function. These in vitro findings suggest that insulin/InsR signaling induces T cell activation with a heightened production of pro-inflammatory cytokines [27]. Similarly, an experiment utilizing a mouse colitis model demonstrated that InsR-deficient T-cells were less effective in inducing chronic colonic inflammation [27].

The expression of InsR was not only observed in T cells but also in other immune cells like tumor-derived macrophage cells [52]. Insulin amplified LPS-mediated cytokine production (TNFα and IL6) in bone marrow-derived macrophages (BMDMs) and human monocytes [53,54]. Chronic insulin treatment was found to enhance macrophage migration towards inflammatory chemokines (MCP-1) from adipose tissue [55]. Moreover, a myeloid lineage-specific *InsR-KO* mouse (*MphIRKO*) exhibited reduced atherosclerotic lesion size in the descending aorta compared to control mice, indicating that insulin/InsR signaling in macrophages promoted obesity-induced atherosclerotic lesion formation [56]. Accordingly, InsR signaling in immune cells has the potential to modulate their inflammatory capacity, highlighting its role in perpetuating the chronic inflammation in T2D. It will be intriguing to investigate whether the interaction between the immune cells and other tissue cells alters their insulin sensitivity, which plays a pathogenic role to propel the cardiovascular complications in T2D.

### 1.2. Association of Adipocyte-Derived Hormone, the Immune System, and T2D-CVDs

Adipose tissue (AT) serves as the site of triglyceride storage and free fatty acid release in response to food availability [57]. Moreover, it functions as an endocrine organ, actively regulating circulating glucose/insulin homeostasis [58]. Increased adipose tissue mass indicates a heightened risk of metabolic disorders such as T2D and its associated complications [59].

Adipokines—bioactive factors secreted by AT—have attracted significant attention. They function as endocrine hormones affecting distant organs or as paracrine signals within local tissues [60]. Among these, adiponectin and leptin are the two significant adipokines, given their contrasting roles in inflammation, insulin sensitivity, and cardiovascular morbidities.

Adiponectin, the most abundant adipokine in plasma, is negatively correlated with T2D and insulin resistance [61]. It exists as three oligomeric complexes, namely a trimer, hexamer, and multimer, with the latter known as high-molecular-weight (HMW) adiponectin [62]. HMW adiponectin can be cleaved by leukocyte elastase to globular adiponectin, the most bioactive isoform [63,64]. Adiponectin exerts biological effects by binding to ADR1 (adiponectin receptor 1), ADR2, or T cadherin [65,66].

Since its discovery in 1995, adiponectin has been shown to exert potent insulin-sensitizing and anti-diabetic effects. Acute injection of purified adiponectin lowers blood glucose in both *WT* (wild-type) and *ob*/*ob* diabetic mice within two hours [67,68]. Similar results were replicated in other independent studies, and continuous infusion of low-dose recombinant adiponectin mitigated hyperglycemia and hyperinsulinemia in HFD (high-fat diet)-fed *WT* mice [61]. Conversely, *adiponectin-KO* mice exhibited severe insulin resistance and elevated serum TNFα following a two-week HFD challenge [69]. These findings collectively suggest that reduced adiponectin levels contribute to the pathogenesis of metabolic disorders in T2D.

Beyond its metabolic effects, adiponectin also plays a protective role in cardiovascular health. Low levels of adiponectin may also be associated with altered immune function in T2D patients, perpetuating chronic inflammation and exacerbating T2D-related cardiovascular complications [70]. For instance, in transgenic mice overexpressing ADR1 (adiponectin receptor 1) in macrophages, body weight is reduced despite unchanged food consumption, indicating that adiponectin has metabolic stimulatory effects through enhanced utilization of glucose and fatty acids in various tissues. Furthermore, *ADR1-TG*/*LdlR-KO* mice showed reduced atherosclerotic lesion size and foam cell infiltration following a HFD challenge [71]. Similarly, the protective effect of adiponectin against atherosclerotic lesion formation was observed in *gAd-TG*/*ApoE-KO* mice (*ApoE-KO* mice, which lack apolipoprotein E, develop atherosclerotic lesions accompanied by hypercholesterolemia), characterized by the overexpression of globular adiponectin [72]. Mechanistically, adiponectin suppresses pro-inflammatory cytokines like TNFα and IL6, consequently mitigating the development of macrovascular morbidities [69,73]. Additionally, by activating PPARγ, adiponectin reduces oxLDL uptake in macrophages and diminishes foam cell formation, highlighting its anti-atherosclerotic effect [74]. Recently, it has been reported that adiponectin, as an immunosuppressor, negatively regulates T cell activation and production of inflammatory cytokines-IFNγ and IL17 [75,76]. Further investigation is imperative to explain whether the deficient adiponectin signaling in immune cells under diabetic conditions unleashes an inflammatory response that exacerbates the progression of T2D-related cardiovascular complications.

In contrast, leptin is another key adipocytokine encoded by the obese (Ob) gene and produced by adipose tissue [77]. Elevated circulating leptin levels have been associated with a higher prevalence of obesity and T2D [78,79,80]. Additionally, hypertensive overweight females have been reported to exhibit higher plasma levels of leptin than their normotensive counterparts [81,82]. Those observations raise the possibility that leptin plays a pathogenic role in T2D-associated CVDs.

Given that atherosclerosis is a major cause of CVDs, researchers have examined the effects of leptin on atherosclerosis in the classic atherosclerotic-prone mouse model. Studies demonstrated that leptin-deficient *ApoE-KO* mice develop smaller atherosclerotic areas compared to leptin-intact *ApoE-KO* mice on a high-fat diet, and exogenous leptin administration increases the atherosclerotic area in *ApoE-KO* mice [83]. Similar findings were reported by other researchers utilizing different atherosclerosis models [83,84,85,86]. One therapeutic study reported that transferring T regulatory cells (Tregs) isolated from *ob*/*ob* mice, which exhibit a significant reduction in IFNγ production, to *ApoE*/*Rag2-KO* (*Rag2-KO* lacks mature adaptive immune cells) mice resulted in a marked decrease in lesion size compared to those receiving *WT*-Tregs [85]. In addition, leptin is known to stimulate T cell proliferation, IFNγ production, glucose uptake, and aerobic glycolysis [28,87,88,89,90,91]. Moreover, leptin acts as a helper cytokine, stimulating M1 polarization and augmenting the production of IL6 and TNFα [92,93]. Leptin-immune activation and its role in the development of T2D and its associated major adverse cardiovascular events represent a promising new area for research (summary of the actions of adipocytokine on immune cells is detailed in Figure 1).

### 1.3. The Dysregulation of Immune Cells in T2D

The over-activation of immune cells has been recognized in T2D patients and preclinical rodent diabetic models [19,94,95]. For example, in adaptive immune cells, flow cytometry evidenced the increased percentage of proinflammatory CD4-derived Th1 and Th17 cells and decreased anti-inflammatory FOXP3 Tregs cells in peripheral blood and adipose tissue in T2D patients [96,97]. Studies using C57BL/6 mice on a high-fat diet revealed increased infiltration of CD8+ T cells into epididymal adipose tissue by immune fluorescent staining, and CD8 depletion parallels with mitigated inflammation [98]. B cells have been reported as a key element in supporting the development of hyperglycemia and insulin resistance in T2D. Mice deficient in B cells showed improved fasting glucose levels and glucose tolerance in T2D mice compared to lean controls [18,99]. Further mechanistic studies indicate that B cell null mice have lowered T cell proinflammatory potential compared to T cells from *WT* obese T2D mice [99].

In innate immune cells, macrophages have received significant attention. Several laboratories reported an increased ratio of proinflammatory M1 macrophages (high TNFα production capacity) to M2 macrophages in the adipose tissue from the diet-induced T2D mouse model and in T2D patients [98,100,101,102]. Macrophage-derived TNFα is the central player in accentuating adipose tissue hyperplasia and death, which in turn facilitates the development of systemic chronic inflammation and metabolic disorders [103]. Natural killer cells, especially the invariant NKT (iNKT), have been reported to be negatively associated with adipose tissue expansion during the development of obesity in patients [104,105]. Adoptive transfer of iNKT cells into obese mice mitigates the metabolic signatures, including fatty livers and insulin resistance, in the diet-induced T2D mouse model [104].

Interestingly, this aberrant overactivation of the immune system does not translate to all immune system function as T2D patients have a higher infection rate than non-diabetic individuals, primarily due to suppressed immune functions that impair pathogen clearance. A clinical study showed that Toll-like receptor-2 (TLR), which serves to detect bacterial and viral pathogen-associated molecular patterns, was reduced in T2D patients with complications [106]. Neutrophils, as standing-by soldiers in the body defending against pathogens, exhibited diminished activity, including reduced ROS and extracellular trap formation in T2D patients [107,108]. Furthermore, macrophages, in response to hyperglycemia, downregulate their Fcγ receptor, resulting in a defect in phagocytosis and antigen presentation while enhancing the production of inflammatory cytokines (e.g., TNFα, IL1β) [109,110]. Another critical immune cell, natural killer cells, responsible for eliminating abnormal or infected cells, also displayed impaired degranulation and activation capacity due to hyperglycemia-induced cellular protein dysfunction in a clinical study [111]. A recent study involving *db*/*db* mice (a classic T2D model lacking leptin receptor, which resembles many aspects of human T2D) infected with *West Nile* virus revealed this paradoxical immune phenotype characterized by heightened inflammatory potential alongside reduced migratory capability [112]. In these mice, cytotoxic CD8T+ cells secreted higher levels of pro-inflammatory cytokines, including IL1, TNFα, and IFNγ while exhibiting impaired migration. In summary, the impaired immune response to pathogenic infection, paralleling chronically inappropriately activated T cells, presents a unique characteristic of T2D patients, which leads to heightened severity and mortality risk associated with infections and cardiovascular events (details summarized in Figure 2). Further therapeutic studies focus on restoring the homeostasis and function of the immune system, which holds the promise to increase both the quality and lifespan of affected patients.

### 1.4. Effects of Cytokines (IFNγ, TNFα, IL17, IL6) on the Development of T2D and Its Related Cardiovascular Complications

Cytokines are exceptionally efficient intercellular communication tools employed by immune cells. They can communicate with distant immune cells to elicit systemic effects, or they can serve as paracrine and autocrine factors to orchestrate localized inflammation [113]. Tumor necrosis factor alpha (TNFα) was the first ‘adipokine’ discovered in the 1980s with multiple functions in both innate and adaptive immunity [114]. Later, it was clarified that TNFα is primarily produced by immune cells such as macrophages [115] and T cells [116]. An elevated level of TNFα was reported in adipose tissue and systemic circulation in obesity and diabetes, contributing to the development of insulin resistance [115,117]. Mechanistic studies have illuminated its role in phosphorylating IRS1 (insulin receptor substrate), converting it into an inhibitor of insulin receptor signaling, leading to insulin resistance in myeloid cells [118]. In addition to its contributing role in insulin resistance, it has been implicated in T2D-related vascular morbidities [119]. TNFα deficiency curtailed atherosclerotic lesion formation, possibly through attenuating foamy macrophages formation and disposition in the arterial wall intima [120,121,122,123]. Anti-TNFα antibody treatment restored the impaired coronary artery dilation in *db*/*db* mice through dampening NFkB (nuclear factor kappa B)-mediated proinflammatory signaling [124]. Additional studies showed that anti-TNFα ameliorated hypertension induced by a high-fat, high-fructose diet in rats [125]. Subsequent mechanistic studies revealed that TNFα impaired the production of NO (nitro oxide) and increased superoxide generation through NADPH oxidase in several vascular beds [122,126,127]. Moreover, elevated TNFα levels are associated with increased immune cell infiltration in kidney specimens from T2D patients with renal disorders [128,129], although TNFα antagonism attenuates hypertension in several models [130]. However, TNFα systemic infusion induces hypotensive and natriuretic effects [130,131]. Future studies are needed to clarify whether and how TNFα-producing immune cells interact with renal and vascular cells, which may provide valuable insight into their role as a potential pathogenic driver of diabetic cardiovascular complications, including heart failure and renal injury.

Interferon-gamma (IFNγ), renowned for its diverse roles in immune regulation, extends its influence beyond traditional immunity, impacting the development of hypertension and contributing significantly to T2D pathogenesis [132]. In T2D, IFNγ stimulated pro-inflammatory macrophage accumulation in adipose tissue (AT), and mice deficient in IFNγ had improved metabolic signatures and decreased numbers of AT macrophages in HFD (high-fat diet)-fed mice [133]. Additionally, IFNγ-neutralizing antibody improved endothelial dysfunction in *db*/*db* mice by curtailing superoxide production [134]. In vitro mechanistic studies reveal that IFNγ enhances oxidative stress and activates endothelial cells and macrophages into pro-inflammatory states, which contribute to metabolic disturbance [135]. Furthermore, an atherogenic role of IFNγ was evidenced by IFNγ-deficient *ApoE-KO* mice, which exhibited a 50% reduction in lesion size and lipid accumulation [136]. Conversely, *ApoE-KO* mice injected with IFNγ showed a 2-fold increase in lesion size and T cell infiltration compared to vehicle injection [137]. Moreover, our research indicates that IFNγ acts as a hypertensive factor by facilitating the formation of immunological synapses between CD8+ T cells and renal tubular cells, leading to increased sodium retention and hypertension in a classic salt-sensitive hypertension model [138]. Given the co-existence of significant elevation in blood pressure and IFNγ in T2D patients [7,139], it is important to investigate whether IFNγ is the “culprit” that contributes to the high prevalence of hypertension among T2D.

Another immune cell-derived cytokine of interest is interleukin 17 (IL17), which modulates immune response by cooperating with other cytokines such as TNFα to amplify the pro-inflammatory signaling in both immune and nonimmune cells [140]. The IL-17 family comprises six members (IL17A to F), and for our purpose, IL17 refers to IL17A here. IL17, also called cytotoxic T-lymphocyte-associated antigen 8 (CTLA-8), is a well-recognized factor to drive an inflammatory cascade through binding to IL-17 receptors [141]. IL17 is mainly produced from the Th17 cells, a subset of CD4+ T cells; however, other immune cells, including CD8+ T cells, γδ T cells, and macrophages, can produce it and are referred to as type 17 cells [142]. Clinically, both obesity and T2D patients have elevated plasma levels of IL17 and an increased number of circulating Th17 cells [143]. Interestingly, the level of IL17 in T2D patients with complications was higher than in patients without complications [144].

IL17, as a pathogenic factor in diabetes, was evidenced by Dr. Ohshima, who showed that anti-IL17 antibodies significantly improved glucose tolerance in *KK-Ay* mice (spontaneous and polygenic T2D model, resembling human T2D features) [145]. Later, Jin reported that IL17-deficient mice were protected against diabetic renal fibrosis, glomerular injury, and albuminuria, and the administration of anti-IL17 antibodies to *WT* diabetic mice showed similar results [146]. In vitro mechanistic studies revealed IL17 and hyperglycemia synergistically upregulated the expression of proinflammatory and pro-fibrotic genes, including IL6, TNFα, and TGFβ, in renal epithelial cells [146]. Beyond its role in precipitating diabetic nephropathy, IL17 signaling was atherogenic, which was supported by smaller lesion formation in both antiIL17 adenovirus-transfected *ApoE-KO* mice [147] and IL17 deficient *LdlR* (low-density lipoprotein receptor)*-KO* mice under Western diet treatment [148]. Further mechanistic studies found IL17 stimulated the production of macrophage-derived pro-inflammatory cytokines (TNFα, IL6, IL1β) and facilitated macrophage infiltration into adipose tissue and the vasculature [149,150,151]. Moreover, IL17 infusion is associated with increased cardiac fibrosis and collagen deposition in the left ventricle [152]. It is also indispensable for AngII-induced blood pressure elevation by activation of renal sodium transporters [153,154]. In summary, IL17 is crucial to promote CVDs in T2D through multiple mechanisms, and future investigation into the factors driving the formation of IL17-producing cells might provide a resolution to mitigate multi-organ injury and cardiovascular comorbidities in T2D.

IL6, an intriguing immune/nonimmune cell-derived inflammatory cytokine [155,156], has been extensively reported to be associated with the occurrence of cardiovascular disease in T2D [157,158]. IL6 executes its signaling through binding to IL6R, which subsequently recruits membrane glycoprotein 130 (gp130) to elicit the classic IL6 signaling cascade [159]. In addition, a complex formed between IL6 and a soluble IL6R (sIL6R) can engage gp130 to trigger IL6 trans-signaling. Due to the broad expression of gp130 across various tissues, IL6 signaling regulates numerous cellular processes, including vascular function, energy homeostasis, and inflammation [160,161].

The metabolic effects of IL6 have been found in *IL6-KO* mice, which exhibit mature obesity, insulin resistance, and hepatic inflammation [162]. A human in vivo study demonstrated that IL6 promotes GLUT4 membrane transportation, as well as enhances lipolysis and fatty acid oxidation in adipose tissue and skeletal muscle cells [163]. Additionally, recent preclinical studies using a high-fat diet mouse model revealed that IL6 infusion improved glucose tolerance, supporting the beneficial metabolic effects of IL6 [164,165].

In the vascular system, however, IL6 exerts detrimental effects, including suppression of insulin-induced NO production and promotion of endothelial activation, and facilitation of immune cell invasion [166,167]. The detrimental vascular effects are primarily attributed to IL6 trans-signaling via sIL6R [168,169]. Interestingly, a naturally occurring soluble form of gp130 (sgp130) acts as an antagonist of sIL6R trans-signaling, maintaining balance in IL6-mediated inflammatory response [170]. Reduced levels of sgp130 have been observed in patients with coronary artery disease [171]. Moreover, inhibition of IL6 trans-signaling by infusion of sgp130 led to significant regression of advanced atherosclerosis and reduction in infiltrating macrophages [172].

Beyond its role in metabolism and vascular biology, IL6 is an instigator to induce pathogenic Th17 cells transformation, which has been known as a major contributor to multiple-organ autoimmunity [173,174]. A recent study using T-cells with constitutive expression of sgp130 reported significant increased Th17 and cytotoxic CD8+ Ts infiltration across multiple organs, coupled with pathophysiological changes in major organs, including the heart and liver [175]. IL6 is also a primary inducer of CRP (C-reactive protein), a well-established prognostic marker in assessing cardiovascular risk in T2D patients [176,177]. However, whether CRP directly contributes to the development of CVD is still controversial [178,179]. Future studies are needed to define if and how IL6 and CRP act synergistically to promote CVD in T2D. In conclusion, IL6 is a double-edged sword, possessing both beneficial metabolic effects and detrimental inflammatory properties. Further research into the cell- and tissue-specific expression of sIL6R and sgp130 is essential to better understand the pro-inflammatory potential of IL6. Such insights may uncover novel therapeutic targets to mitigate localized chronic inflammation in T2D and its cardiovascular complications.

Collectively, cytokines serve as critical mediators in both the induction and progression of T2D and its associated cardiovascular complications (summarized in Figure 3). Understanding their complex roles and interactions is essential for developing targeted therapeutic interventions to address these multifaceted pathologies in T2D [180,181].

## 2. Summary/Perspective

Given that obesity-induced T2D represents a metabolic disorder characterized by disturbed hormone balance, the current review examines the action of insulin, adiponectin, and leptin in immune activation, which contributes to the aberrant activity of immune cells in T2D [182,183,184]. Both insulin and leptin enhance the pro-inflammatory properties of T cells and macrophages [90,185], whereas adiponectin functions as an immune suppressor by limiting T cell activation and IFNγ production [76,186]. The diabetic milieu, characterized by elevated insulin and leptin levels alongside reduced adiponectin, thus fosters a chronic inflammatory environment via enhancing the production of pro-inflammatory cytokines from diverse immune cells (Figure 1). Notably, cytokines IFNγ, TNFα, IL6, and IL17, released from immune cells, have emerged as critical pathogenic mediators in metabolic disorder and vascular morbidities in T2D (Figure 2) [181,187]. However, the lack of immune cell-specific knockout mice remains a limitation in fully elucidating the pathogenic role of immune cells in precipitation of cardiovascular complications in T2D. Therefore, future studies employing T cell-specific knockout mice will be essential to reveal the unique role of immune cells in sustaining chronic inflammation and driving tissue dysfunction, which ultimately contribute to multiple complications commonly observed in T2D.

Furthermore, the interaction between the individual cytokines can amplify their deleterious metabolic and cardiovascular effect. For example, IL6 is an indispensable cytokine to stimulate IL17 production through inducing Th17 polarization, while IFNγ can activate macrophages and stimulate TNFα and IL6 secretion. Therefore, a combinatory therapeutic approach that inhibits multiple cytokine-signaling holds promising potential to decouple cardiovascular disease from type 2 diabetes (T2D). However, the redundancy in cytokine signaling might limit the long-term efficacy of cytokine therapy, as the immune system can quickly compensate to counteract the initial anti-inflammatory effects.

Another promising therapeutic approach involves targeting the unique signature presented on T2D-immune cells, enabling recognition by CAR-T cells to induce apoptosis and mitigate multiple pathogenic mechanisms driven by dysfunctional immune cells. Additionally, new treatments that activate our self-anti-inflammatory mechanisms, such as promoting Treg cell differentiation [188,189] —may restrain the overactivation of immune cells and thereby attenuate the development of cardiovascular complications in T2D (new potential therapies are elaborated in Figure 4).

Notably, obesity-induced T2D does not develop overnight; investigating factors present in the obese prediabetic state, such as hyperinsulinemia and hyperleptinemia, may reveal critical mechanisms driving the accumulation and infiltration of immune cells. These insights could provide new preventative therapies with the hope of disconnecting cardiovascular morbidities in T2D.

## Figures and Tables

**Figure 1 life-15-01241-f001:**
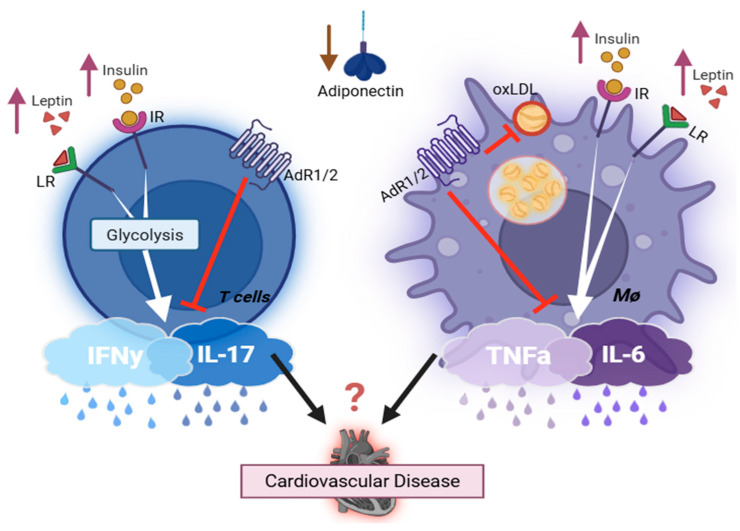
Key endocrine hormones involved in regulating the immune cell’s inflammatory capacity. Created with BioRender.com. Abbreviations: IR = insulin receptor; LR = leptin receptor; AdR1/2 = adiponectin receptor; oxLDL = oxidized low-density lipoprotein cholesterol; L/M/HMW = low/medium/high molecular weight. **Leptin and insulin**, both elevated in obesity T2D, enhance pro-inflammatory cytokine production in immune cells (TNFα, IL6, IFNγ, IL17), thereby exacerbating cardiovascular inflammation and disease progression. **Adiponectin** is abundantly secreted by adipocytes but markedly reduced in obesity type 2 diabetes. It exerts anti-inflammatory effects via suppressing pro-inflammatory mediators from both T cells and macrophages. Additionally, it inhibits oxidized LDL uptake by macrophages, reducing foam cell formation and lowering the risk of cardiovascular diseases. **Adiponectin complexes**: Circulating forms include trimer (LMW), hexamer (MMW), HMW multimers (12–18 mer), and the globular form derived from proteolytic cleavage of the full-length protein. The HMW form correlates most strongly with insulin sensitivity and metabolic health, and globular adiponectin has particularly potent anti-inflammatory effects.

**Figure 2 life-15-01241-f002:**
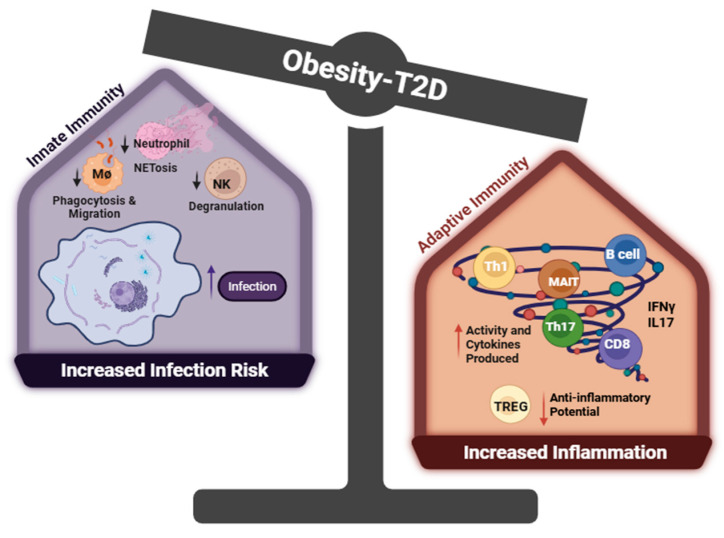
Dysfunction of the immune system in T2D. Created with BioRender.com. Abbreviations: Th1 = CD4+ T cells with high IFNγ production, Th17 = CD4+ T Cells with high IL17 production, Treg = T Regulatory CD4+ T cells, NK = nature killer cells, MØ = macrophage, MAIT = mucosal-associated invariant T cell. In obesity-associated type 2 diabetes (T2D), innate immune cells exhibit functional impairments, including reduced neutrophil extracellular trap (NET) formation, diminished natural killer (NK) cell degranulation, and impaired macrophage phagocytosis—all of which are associated with increased infection risk and mortality. Within the adaptive immune compartment, regulatory T cells (Tregs)—critical for maintaining immune tolerance and resolving inflammation—are significantly reduced in obesity and T2D, compromising their suppressive function. In contrast, proinflammatory subsets such as Th1, Th17, and cytotoxic CD8+ T cells are highly activated: Th1 and CD8+ T cells secrete IFN-γ to drive M1 macrophage polarization and promote adipose tissue inflammation, while Th17 and MAIT cells produce IL17, further exacerbating systemic inflammation.

**Figure 3 life-15-01241-f003:**
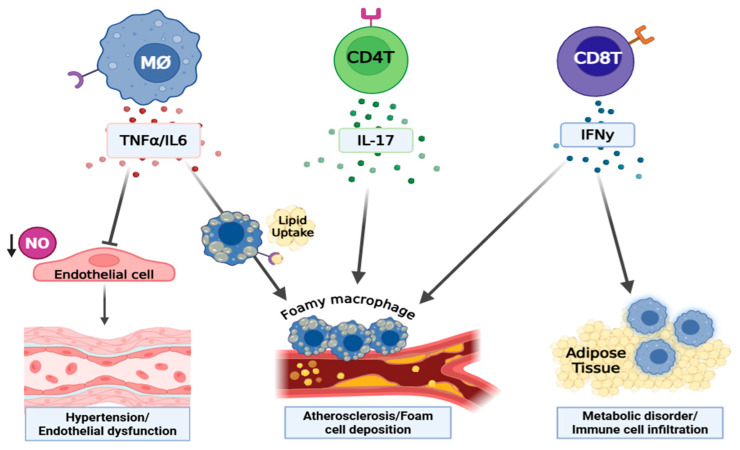
The involvement of immune cell-derived cytokines in the development of T2D and its vascular complications. Created by BioRender.com. Abbreviations: NO = nitric oxide, MØ = macrophage, M1Ø = type1 macrophage, CD4+ Ts = CD4+ lymphocytes. Macrophage-derived TNF-α impairs endothelial function by suppressing NO production and increasing superoxide production (via NADPH oxidase) that promotes vasoconstriction, vascular injury, and hypertension. IL6, also produced by macrophages, contributes to endothelial dysfunction and vascular inflammation primarily through trans-signaling: IL6 binds soluble IL6 receptor (sIL-6R), forming a complex that engages gp130 on cells lacking membrane-bound IL-6R, thereby driving pro-inflammatory cascades across many cell types. This IL6 trans-signaling axis is strongly associated with chronic vascular injury, atherosclerosis, and hypertension. IFN-γ, secreted by T-cells, enhances inflammation in adipose tissue by promoting M1 polarization of macrophages, exacerbates metabolic dysfunction, and causes endothelial injury. IL-17, produced predominantly by Th17 cells, promotes atherogenesis through enhancing the expression of pro-inflammatory chemokines, thereby driving monocyte/macrophage recruitment into vascular lesions and fibrotic sites. This leads to aggravated vascular inflammation and lesion progression.

**Figure 4 life-15-01241-f004:**
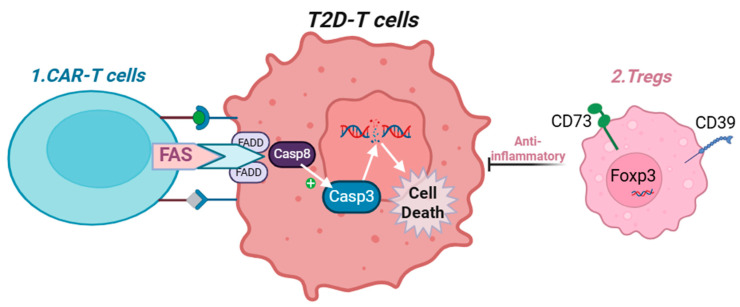
Proposed mechanism to suppress T cell activity in T2D. Created with BioRender.com. Abbreviations: Treg = T Regulatory CD4+ T cells, FADD = Fas-associated death domain, Casp8 = caspase 8, Foxp3 = Forkhead box P3. CAR-T = chimeric antigen receptor T cells. **1**: Chimeric antigen receptor (CAR) T cells engineered to express Fas ligand (FasL) can recognize unique surface markers on T2D-associated T cells and induce their apoptosis. **2**: Overexpression of CD39 and CD73 promotes the differentiation of regulatory T cells (Tregs), which possess strong anti-inflammatory properties and can suppress the activity of activated immune cells in T2D.

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
