# Peer review of "Immune Dysregulation Connecting Type 2 Diabetes and Cardiovascular Complications"

_life, 2025, doi:10.3390/life15081241_

Round 1

Reviewer 1 Report

Comments and Suggestions for Authors

Title:  Immune Dysregulation Connecting Type 2 Diabetes and 2 Cardiovascular Complications

This review article aims to focus on disrupted homeostasis, a characteristic of diabetic conditions, encompassing cytokines, hormones, and their interplay with changes in immune function. Comments related to the manuscript are mentioned below.

  1. The introduction section is informative and well-cited.
  2. Authors are advised to add two more figures: first, elaborating on the dysregulation of immune cells in type 2 diabetes mellitus at the molecular level, and second, on the mechanism of immune dysfunction associated with type 2 diabetes mellitus. This will enhance the readability of the manuscript.
  3. The literature review is interesting to read. Still, since an abundance of literature is available on this aspect, authors should elaborate pointwise on how this differs from other manuscripts and what the novelty of this article is.
  4. Additionally, the authors should explain how this review article will benefit the scientific community and what knowledge it will contribute to the existing understanding of this aspect.
  5. They should elaborate on the gaps in knowledge that need to be explored in the area of “Immune Dysregulation Connecting Type 2 Diabetes.” This can be added to the conclusion section.

Author Response

Comments 1: The introduction section is informative and well-cited.

Response 1:
The reviewer found the introduction informative and well-cited; therefore, no changes were made to this section.

Comments 2: Authors are advised to add two more figures: first, elaborating on the dysregulation of immune cells in type 2 diabetes mellitus at the molecular level, and second, on the mechanism of immune dysfunction associated with type 2 diabetes mellitus. This will enhance the readability of the manuscript.

Response 2: We created two new figures to enhance clarity. Figure 1 illustrates the molecular mechanisms by which insulin, leptin, and adiponectin contribute to immune cell dysfunction in T2D. Figure 3 outlines the mechanisms of immune dysregulation, including aberrant activation of adaptive immune cells, heightened cytokine production, and suppressed innate immune cell function in T2D.

Comments 3:

The literature review is interesting to read. Still, since an abundance of literature is available on this aspect, authors should elaborate pointwise on how this differs from other manuscripts and what the novelty of this article is.

Response 3: We added an entirely new discussion in Section 2 to highlight our unique perspective: that the imbalance of endocrine hormones—specifically elevated insulin and leptin and reduced adiponectin—drives immune cell dysfunction, contributing to the pathogenesis of T2D-associated cardiovascular disease (T2D-CVD).

Comments 4: Additionally, the authors should explain how this review article will benefit the scientific community and what knowledge it will contribute to the existing understanding of this aspect.

Response 4: To benefit the scientific community, we propose novel therapeutic strategies in paragraphs 2 and 3 of Section 2, aiming to guide future research directions.

Comments 5: They should elaborate on the gaps in knowledge that need to be explored in the area of “Immune Dysregulation Connecting Type 2 Diabetes.” This can be added to the conclusion section.

Response 5: We addressed the knowledge gap in conclusion (Section 2, first paragraph), noting that a major gap in understanding immune cell involvement in T2D lies in the lack of T cell-specific knockout models.

Reviewer 2 Report

Comments and Suggestions for Authors

The review is on the diabetes and cardiovascular complications associated with it.

Specific points:

1. Individual authors are not connected to the affiliation. Even if all the authors are affiliated with the one mentioned institution, who is the corresponding author? These elements are part of the standard template.

2. Keywords are also provided in the template, although they are not critical for the initial evaluation of the manuscript.

3. Something is wrong with the reference list. After reference 24 the next one is 69, while it is just the beginning of the review (line 45, page 2).

4.  There are many abbreviations, including non-standard and field-specific. It would be useful for readers to reduce the number of abbreviations used, and to provide a list of abbreviations, as other MDPI review articles often do. 

5. Db/Db or db/db mice are mentioned three times, it needs to be introduced in more detail when mentioned first time. 

6. All other mentioned mice (ApoE-KO, Rag2, etc) need to be properly introduced with short descriptions, and knockout mouse names need to be provided using established standards, e.g., italics, superscripts, genes modified, at least when introducing the mice.

7. Latin names are usually italicized, such as S. aureus (line 282, but check the entire manuscript to be sure).

8. Consider avoiding long paragraphs such as lines 264-291, to make the article easier to read.

9. Acknowledgements, kindly follow the MDPI rules for it, and also consider if they can be co-authors instead by contributing to the manuscript, unlike all other co-authors, whose contribution is not mentioned.

10. Lines 332-333 on ethics can be removed for the review article.

11. Line 339 for data availability can be removed for the Review article: there are no data by definition.

12. Author contributions. Only one author (Y.L.) wrote the manuscript and designed the Figure. All other authors did not contribute and should not be listed as co-authors. 

13. The reference list needs to be updated. While there are only 8/158 references for 2023 (including some review articles), there are none for 2024 and 2025. For older references, it is suggested to use original research articles. Old reviewers are included exceptionally.

Author Response

We are grateful for the reviewer’s comment, which contributed to improving the overall clarity of the paper.

Comment 1: Individual authors are not connected to the affiliation. Even if all the authors are affiliated with the one mentioned institution, who is the corresponding author? These elements are part of the standard template.

Response 1: All authors are from the same institute. An asterisk has been added to indicate the corresponding/last author.

Comments 2: Keywords are also provided in the template, although they are not critical for the initial evaluation of the manuscript.

Response 2: Keywords have been added following the abstract.

Comments 3: Something is wrong with the reference list. After reference 24 the next one is 69, while it is just the beginning of the review (line 45, page 2).

Response 3: Citation numbering errors have been corrected.

Comments 4: There are many abbreviations, including non-standard and field specific. It would be useful for readers to reduce the number of abbreviations used, and to provide a list of abbreviations, as other MDPI review articles often do. 

Response 4: All abbreviations used in the four figures have been expanded within the figure legends. In addition, an abbreviation list has been added after the keywords section.

Comments 5: Db/Db or db/db mice are mentioned three times, it needs to be introduced in more detail when mentioned first time. 

Response 5: A brief introduction to the db/db mouse model has been added at its first mention (line 244).

Comments 6: All other mentioned mice (ApoE-KO, Rag2, etc) need to be properly introduced with short descriptions, and knockout mouse names need to be provided using established standards, e.g., italics, superscripts, genes modified, at least when introducing the mice.

Response 6: Short introductions for ApoE-KO, Rag2-KO and other transgenic mice have been added at their first appearance in the text and italicized.

Comments 7: Latin names are usually italicized, such as S. aureus (line 282, but check the entire manuscript to be sure).

Response 7: All Latin species names have been italicized.

Comments 8: Consider avoiding long paragraphs such as lines 264-291, to make the article easier to read.

Response 8: Lines 264–291 were revised for clarity and conciseness. The simplified content now appears as the third paragraph of Section 1.3 (lines 221–244).

Comments 9: Acknowledgements, kindly follow the MDPI rules for it, and also consider if they can be co-authors instead by contributing to the manuscript, unlike all other co-authors, contribution is not mentioned.

Response 9: All previously acknowledged contributors have now been listed as co-authors, and individual contributions have been updated accordingly.

Comments 10: Lines 332-333 on ethics can be removed for the review article.

Response 10: The ethics statement (lines 332–333) has been removed.

Comments 11: Line 339 for data availability can be removed for the Review article: there are no data by definition.

Response 11: The data availability statement (line 339) has also been removed.

Comments 12: Author contributions. Only one author (Y.L.) wrote the manuscript and designed the Figure. All other authors did not contribute and should not be listed as co-authors. 

Response 12: Author contribution statements have been updated to reflect each individual's role.

Comments 13: The reference list needs to be updated. While there are only 8/158 references for 2023 (including some review articles), there are none for 2024 and 2025. For older references, it is suggested to use original research articles. Old reviewers are included exceptionally.

Response 13: 

We appreciate the reviewer’s suggestion. Two recent references from 2024 (References 34 and 174) have now been added to incorporate the latest research developments. In addition, older references have been reviewed and updated, with a preference given to original research articles. Older review articles have been retained only when they provide exceptional and foundational insight.

Reviewer 3 Report

Comments and Suggestions for Authors

Deck K et al., reviewed the role of key immune components that are implicated in T2D and T2D-associated CVD, with in-depth discussions. However, current version of manuscript is hard to follow. Specific comments include:

  1. The lengthy introduction section should be break into several sections, with each discussing different aspects. For example, introduction can be follow by sections that discuss: (1) role of immune system components in T2D; (2)  T2D-associated inflammation that elevates CV risk, (3) adaptive immune system's role in elevated CV risk in T2D patients etc.
  2. Flow in 1.1 needs optimization. Authors went into detailed discussion on T cells and macrophages in line 57-74, which is followed by high level evidence of how Ins/InsR could promote CVD in T2D. Given the topic of the review, fourth paragraph should, instead, be the first paragraph in this section, followed by overall of Ins/InsR in T2D, Ins/InsR function in T cells, Ins/InsR function in Macrophages.
  3. Section 1.2 should be broken into 3 subsections: overview (line 86-98), discussions on adiponection (line 99-130), and leptin discussions (131-151).
  4. It is not clear why discussions in 1.3 are at high level, contrasting its potential paramount importance to the topic of immune dysregulation in T2D-associated CV risk
  5. Authors should consider reorganize paragraph in line 185-227, and highlighting discussion on how each of the TNF/IFNg links T2D to elevated CV risks.
  6. It's unclear why section 1.5 is included, as its discussions around increased infection risk doesn't seem to be coherent to the thesis (CVD in T2D).
  7. IL-6 and CRP are both important in both T2D and CVD (Ref 1-3), and authors should consider add relevant sections reviewing their roles in T2D-associated CVD risk.

Overall, authors provided in-depth review on immune system's role in T2D and T2D associated CVD, but the manuscripts requires substantial revision before being published.

Refs:

  1. Tian R et al., (2019) C-reactive protein for predicting cardiovascular and all-cause mortality in type 2 diabetic patients: A meta-analysis. Cytokine. 117:59-64
  2. Qu D et al., (2014). IL-6 in diabetes and cardiovascular complications. Br J Pharmacol. 171(15):3595-3603
  3. Pradhan AD et al., (2001). C-Reactive Protein, Interleukin 6, and Risk of Developing Type 2 Diabetes Mellitus. JAMA. 286(3): 327-334

Author Response

We are grateful for the reviewer’s comment, which contributed to improving the overall clarity of the paper.

Comments 1: The lengthy introduction section should be break into several sections, with each discussing different aspects. For example, introduction can be follow by sections that discuss: (1) role of immune system components in T2D; (2)  T2D-associated inflammation that elevates CV risk, (3) adaptive immune system's role in elevated CV risk in T2D patients etc.

Response 1: We thank the reviewer for this constructive suggestion. The introduction has been reorganized and divided into three distinct paragraphs to improve clarity and structure. The first paragraph outlines the prevalence and significance of cardiovascular disease (CVD) in the context of type 2 diabetes (T2D). The second highlights the need to explore additional, yet unidentified, pathogenic factors that may help decouple CVD from T2D. A third paragraph has been added to introduce the emerging recognition of the immune system and cytokines as key contributors to T2D-associated CVD. Additionally, we introduce the concept that metabolic hormone receptor signaling in immune cells may represent an underexplored pathogenic pathway, potentially driving immune activation and contributing to CVD progression in T2D.

Comments 2: Flow in 1.1 needs optimization. Authors went into detailed discussion on T cells and macrophages in line 57-74, which is followed by high level evidence of how Ins/InsR could promote CVD in T2D. Given the topic of the review, fourth paragraph should, instead, be the first paragraph in this section, followed by overall of Ins/InsR in T2D, Ins/InsR function in T cells, Ins/InsR function in Macrophages.

Response 2: The flow of Section 1.1 has been reorganized per the reviewer’s suggestion. The content previously in the last paragraph now appears at the beginning of the section.

Comments 3: Section 1.2 should be broken into 3 subsections: overview (line 86-98), discussions on adiponection (line 99-130), and leptin discussions (131-151).

Response 3: Section 1.2 has been restructured into three major subsections based on the reviewer’s suggestion, comprising seven paragraphs:

  • Paragraphs 1–2 introduce adipose tissue and adipokines.
  • Paragraphs 3–5 focus on adiponectin.
  • Paragraphs 6–7 discuss leptin.

Comments 4: It is not clear why discussions in 1.3 are at high level, contrasting its potential paramount importance to the topic of immune dysregulation in T2D-associated CV risk

Response 4: We appreciate the reviewer’s observation and agree that Section 1.3 is critical to the overall framework of the review. While immune dysregulation in T2D has been extensively discussed in the literature, our aim was to present a concise yet distinctive overview that emphasizes two underappreciated aspects: (1) chronic immune activation, largely driven by heightened adaptive immune cell activity, and (2) suppressed innate immune responses to pathogenic insults—both of which are relevant to increased mortality in T2D. To address concerns about clarity and depth, we have added a new figure summarizing these key features of immune dysfunction in T2D. This section also functions as a conceptual bridge to the following section, which delves into cytokines—the key immune cell-derived mediators contributing to T2D-associated cardiovascular complications.

Comments 5: Authors should consider reorganize paragraph in line 185-227, and highlighting discussion on how each of the TNF/IFNg links T2D to elevated CV risks.

Response 5:
Lines 183–227 have been revised and reorganized. The discussion of TNF and IFNγ has been separated into distinct sections to improve clarity and reader engagement.

Comments 6: It's unclear why section 1.5 is included, as its discussions around increased infection risk doesn't seem to be coherent to the thesis (CVD in T2D).

Response 6:
We appreciate the reviewer’s feedback. In response, Section 1.5 has been simplified and integrated into the third paragraph of Section 1.3 to improve coherence. The revised content emphasizes impaired innate immune function in T2D and its relevance to increased infection risk, while maintaining focus on the broader theme of immune dysregulation in T2D, which underlies both infection susceptibility and cardiovascular complications.

Comments 7: IL-6 and CRP are both important in both T2D and CVD (Ref 1-3), and authors should consider add relevant sections reviewing their roles in T2D-associated CVD risk.

 Response 7:
A new paragraph has been added to Section 1.4 (fourth paragraph), discussing IL-6 and CRP and their association with T2D and related cardiovascular complications. All three reviewer-recommended references have been incorporated.

Reviewer 4 Report

Comments and Suggestions for Authors

The manuscript titled “Immune Dysregulation Connecting Type 2 Diabetes and Cardiovascular Complications” provides a detailed review of how immune dysfunction contributes to cardiovascular disease (CVD) in the context of type 2 diabetes (T2D). The paper is comprehensive and scientifically grounded, but there are several areas where it could be strengthened.   1. The summary in Figure 1 is too high-level and does not include most of the points discussed in this review (like insulin and other hormones discussed in the manuscript).   2. In section 1.1, the authors introduced insulin. How about insulin resistance? The authors should provide a deeper review of insulin resistance and how changes in insulin signaling lead to T2D and cardiovascular complications.   3. Terms like “T2D patients” vs. “diabetic patients” and “WT” vs. “wild-type” should be standardized.   4. In lines 19–22, the authors cited 2019 data on diabetes. It is 2025 now; please update as 'latest estimates'.

Author Response

Answers to reviewer 4:

We thank the reviewer for their comment, which improved the paper’s clarity.

Comments 1: The summary in Figure 1 is too high-level and does not include most of the points discussed in this review (like insulin and other hormones discussed in the manuscript).

Response 1:
We appreciate the reviewer’s comment and have revised the figures accordingly to better align with the content of the review. Specifically, we have added three new figures (Figures 1, 2, and 4) and updated the original Figure 1, now presented as Figure 3. The new Figure 1 illustrates the roles of insulin and adipokines in regulating immune cell activity. Figure 2 summarizes immune cell dysfunction in T2D. Figure 3 depicts the mechanisms by which cytokines contribute to the development of cardiovascular disease (CVD) in T2D. Lastly, Figure 4 highlights emerging therapeutic strategies targeting aberrantly activated immune cells in T2D.

Comments 2: In section 1.1, the authors introduced insulin. How about insulin resistance? The authors should provide a deeper review of insulin resistance and how changes in insulin signaling lead to T2D and cardiovascular complications.  

Response 2:
We appreciate the reviewer’s suggestion. An expanded discussion on insulin resistance in T2D has been added to the first paragraph of Section 1.1 (lines 77–85), providing more detail on how “insulin resistance”, including both retained and suppressed insulin signaling, contributes to disease pathogenesis. The cardiovascular effects of insulin signaling are now addressed in the second paragraph (lines 86–94), and we have also highlighted the role of insulin signaling in T cells as a potential immune-mediated mechanism promoting CVD in T2D, discussed in the third paragraph of the same section.

Comments 3: Terms like “T2D patients” vs. “diabetic patients” and “WT” vs. “wild-type” should be standardized.

Response 3:
We have clarified the distinction between T2D patients and diabetic patients. The term diabetic patients encompass individuals with all forms of diabetes, whereas T2D patients refers specifically to those with type 2 diabetes. In the revised manuscript, all data specifically pertaining to type 2 diabetes are consistently labeled as T2D patients. The term diabetic patients is retained in line 275, as the referenced study includes both type 1 and type 2 diabetes. The abbreviation WT has also been standardized throughout the manuscript.

Comments 4: In lines 19–22, the authors cited 2019 data on diabetes. It is 2025 now; please update as 'latest estimates'.

Response 4:
Thank you for pointing this out. The most recent estimates of diabetes prevalence, based on 2024 data, have been incorporated and are now reflected in line 33 of the revised manuscript (Line 33).

Round 2

Reviewer 1 Report

Comments and Suggestions for Authors

Authors have worked hard to improve the manuscript. They have incorporated all the suggestions. Manuscript may be accepted for publication.

Reviewer 2 Report

Comments and Suggestions for Authors

The manuscript has been revised according to the reviewer's comments.

The author contribution has been added and expanded, although not by the MDPI template standards.

Some references were added, although the majority of references are still outside the last 5 years. 

Reviewer 3 Report

Comments and Suggestions for Authors

Thanks for authors' efforts in addressing my comments. After the revision, I believe the manuscript is in good shape for publication.